# An Improved Vulnerability Exploitation Prediction Model with Novel Cost Function and Custom Trained Word Vector Embedding

**DOI:** 10.3390/s21124220

**Published:** 2021-06-20

**Authors:** Mohammad Shamsul Hoque, Norziana Jamil, Nowshad Amin, Kwok-Yan Lam

**Affiliations:** 1College of Computing & Informatics, Universiti Tenaga Nasional, Kajang 43000, Malaysia; Norziana@uniten.edu.my; 2Renewable Energy and Solar Photovoltaics, Institute of Sustainable Energy (ISE), Universiti Tenaga Nasional, Kajang 43000, Malaysia; Nowshad@uniten.edu.my; 3Technopreneur-Ship Centre, School of Computer Science and Engineering and Director of the Nanyang, Nanyang Technological University (NTU), Singapore 639798, Singapore; kwokyan.lam@ntu.edu.sg

**Keywords:** cloud security management, supervised machine learning, modelling and prediction, cost function, vulnerability exploitation prediction

## Abstract

Successful cyber-attacks are caused by the exploitation of some vulnerabilities in the software and/or hardware that exist in systems deployed in premises or the cloud. Although hundreds of vulnerabilities are discovered every year, only a small fraction of them actually become exploited, thereby there exists a severe class imbalance between the number of exploited and non-exploited vulnerabilities. The open source national vulnerability database, the largest repository to index and maintain all known vulnerabilities, assigns a unique identifier to each vulnerability. Each registered vulnerability also gets a severity score based on the impact it might inflict upon if compromised. Recent research works showed that the cvss score is not the only factor to select a vulnerability for exploitation, and other attributes in the national vulnerability database can be effectively utilized as predictive feature to predict the most exploitable vulnerabilities. Since cybersecurity management is highly resource savvy, organizations such as cloud systems will benefit when the most likely exploitable vulnerabilities that exist in their system software or hardware can be predicted with as much accuracy and reliability as possible, to best utilize the available resources to fix those first. Various existing research works have developed vulnerability exploitation prediction models by addressing the existing class imbalance based on algorithmic and artificial data resampling techniques but still suffer greatly from the overfitting problem to the major class rendering them practically unreliable. In this research, we have designed a novel cost function feature to address the existing class imbalance. We also have utilized the available large text corpus in the extracted dataset to develop a custom-trained word vector that can better capture the context of the local text data for utilization as an embedded layer in neural networks. Our developed vulnerability exploitation prediction models powered by a novel cost function and custom-trained word vector have achieved very high overall performance metrics for accuracy, precision, recall, F1-Score and AUC score with values of 0.92, 0.89, 0.98, 0.94 and 0.97, respectively, thereby outperforming any existing models while successfully overcoming the existing overfitting problem for class imbalance.

## 1. Introduction

Vulnerability reflects the weakness of a system for which most of the security takes place. Since there is a growth in the vulnerability trend and so does the threat of security breach [1], the importance for security focus has therefore dramatically increased. Various cloud service models provide computing platforms that enable consumers to perform many tasks; today, organizations and governments are shifting more and more computational workloads to the cloud. As organizations continue to develop new applications or migrate existing applications to cloud-based services, their concerns for safe computational environments become top priority. While the concern is meaningful, the security management—when cloud computing security is performed correctly—can be as safe as traditional on premise IT.

This research work has objective to enhance the cost effective security patching of any software, hardware or systems deployed in cloud or in-house environment and security vulnerabilities published in the national vulnerability database are considered in the research scope. Although a large number of cyber vulnerabilities are discovered every year, yet only a small number are reported to be exploited, except for unreported and unknown exploitations in the wild. According to the 2019 Mid-Year VulnDB QuickView report [2], 10,644 vulnerabilities have been published with Common Vulnerabilities and Exposures (CVE) identifiers and only about 10% of those have confirmed PoC (proof of Concept) exploits attached to them. Security patching is a costly, broad and difficult operational work that involves triages for vulnerabilities, and the most critical vulnerabilities should be patched first. The CVSS score has become a common tool by the industry standard for assessing the severity of a vulnerability. However, previous studies have shown that the CVSS score may not be the ideal tool to determine which vulnerability should be patched first, since a higher CVSS score is not the determinant for exploitation likelihood [3]. So, it is important to predict the vulnerabilities that are most likely to get exploited, to avoid damages of critical assets and businesses, especially for systems that are designed and deployed without assuming access to proprietary information.

When the most exploitable vulnerabilities are known, significant security management decisions can be made proactively to prioritize and deploy available resources for security inspection, testing, and patching accordingly [4,5,6,7,8,9]. Although research in vulnerability prediction has been a popular topic for years, yet it has only recently achieved stronger traction in the industry [10]. Researchers have used vulnerability databases of various forms to develop models for vulnerability disclosure trends. Most of this research is aimed at finding techniques in developing models that can predict the most likely exploitable vulnerabilities of products in the future by using existing features such as text data from a summary field or various sources of the social media [10,11,12,13,14,15,16,17,18]. For these models, predicting the most exploitable vulnerabilities with supervised classification-based machine learning algorithms, the target label (binary class) has a severe imbalance (approximately 1 to 12) in favour of the major class (number of non-exploited vulnerabilities) since only a small fraction of published vulnerabilities have validated exploitation codes available in exploit databases.

The vulnerability exploitation prediction models in existing research works have tried to address this class imbalance issue with artificial data resampling and algorithmic approaches which have limitations stemming from their underlying assumptions through the usage of machine learning algorithms, application techniques, dataset and feature set selections, as described in [13]. These approaches have increased the model performances slightly but still suffered greatly from model overfitting problem (high accuracy but low precision, recall etc.). In this research work, this issue is addressed with most advanced option of developing an appropriate cost function which does not work by artificial data resampling and algorithmic approach but can address the existing class imbalance by providing higher weightage to the minor class and vice versa. The cost function was designed out of data engineering from existing attribute (CWE_ID) of the extracted dataset and utilized as predictive feature which effectively eliminates overfitting issue while greatly improved the overall model performance metrics such as accuracy, precision, recall, F1-score and AUC score. The extracted dataset has a large text corpus in the summary attribute which is suitable to create a custom trained word since it can better capture the context of the local text and therefore this research has also developed a custom trained word vector instead of using popular pre-trained word vectors such as ‘GloVe (FastText)’ and ‘wiki-news-300d-1M.vec (Word2vec) to utilize as an embedded layer in the neural networks This utilization of custom trained word vector also contributed to performance increase for neural models since it could better capture the context of the local text data. The security activities for software and systems are highly resource savvy, it is thus expected that the models can predict vulnerabilities which are most likely to be exploited with high accuracy and reliability. This yields the need in exploring additional models as expected by the vendors, end users and businesses [18,19,20]. The aim of this research work is to develop binary classification based machine learning models capable of performing, with improved accuracy, precision, recall and F1 score.

The contribution of this research work is three-fold, as described below:Design and development of a novel cost function (named as ‘co_efficient_balance’) to address the existing class imbalance in the extracted dataset. The cost function is able to aid the classification algorithms by efficiently differentiating the binary classes and thereby to predict the label classes with overall great performance in accuracy, precision, recall and F1-score.Development of a custom-trained word vector out of the text corpus from the ‘summary’ field of the extracted dataset (termed as ‘vulnerability.vec’) to be utilized as the embedded word feature layer in the neural network models. This custom-trained word vector has outperformed popular pre-trained word vectors such as ‘GloVe (FastText)’ and ‘wiki-news-300d-1M.vec (Word2vec)’.The proposed exploitation prediction model which have utilized the novel cost function (co_efficient_balance) and text data embedded by the custom-trained word vector (vulnerability.vec) as features for classification algorithm with long short-term neural network (LSTM), a variant of the recurrent neural network (RNN) have achieved great overall performance scores in accuracy, precision, recall, F1-Score and AUC score values of 0.92, 0.89, 0.98, 0.94 and 0.97 respectively, these performances are better than any existing such models.

## 2. Related Research

### 2.1. Existing Vulnerability Exploitation Prediction Models

Almukaynizi et al. [11] developed a cyber-vulnerability prediction model to predict whether a vulnerability threat will be exploited or not. Data were collected from NDV, Exploit DB, Zero-day initiative (ZDI) and marketplaces of Dark and Deep web forums. They found that the use of D2W data increased the prediction capability and that texts related to vulnerability in the Russian language had greater possibility to get exploited. The union of the Symantec’s anti-virus attack signatures and the Intrusion Detection Systems’ (IDS) attack signatures were considered as ground truths and text data processed by the TF-IDF model were found as the most important prediction feature. Although their best model achieved a high True Positive Rate and low False Positive Rate (90%, 13%, respectively), the overall F1 score was only 0.40 which might be caused by overfitting to the major class.

Nazgol Tavabi et al. [12] developed another vulnerability exploitation prediction model with word embedding (termed as “Dark Embed”). The text corpus was extracted form Dark and deep web posts and discussions mentioning vulnerability. “CVSS” scoring and the presence of vulnerability in Exploit DB were utilized as features for prediction while Symantec’s anti-virus and Intrusion Detection Systems attack signatures were used as the ground truth. Their model achieved an F1 score of 0.74. No information was provided either on other performance metrics, such as accuracy, precision and recall or on whether the custom-created word embedding performed better than popular-pertained word vectors such as glove or wiki.300d.vec.

Edkrantz et al. [13] utilized the SVM classier to develop an exploitation prediction model for vulnerability using the NVD dataset and a commercial dataset from Recorded Future as feature set and Exploit DB as the ground truth. The Commercial firm Recorded Future harvests data from various sources such as Twitter, Facebook, blogs, news articles, and RSS feeds. Their experiments found text features to be the most important predictive, and the best model using the LibSVM algorithm achieved an overall good performance, such as accuracy, recall and F1 score of 0.83, 0.82 and 0.83 respectively. 

Reinthal et al. [14] developed two vulnerability exploitation prediction models (naïve and realistic). The naïve model used temporal intermixing while the realistic model tried to avoid that. A combination of the NVD and web chatter data was utilized as feature dataset while the exploit DB as the ground truth. They have also found textual data modelled with TF-ITF as the most important feature for prediction. Although Xgboost, a supervised ML algorithm, was used to increase the result metrics of the imbalanced dataset, their best model (realistic) was able to achieve low values of precision, recall and F1 score of 0.578, 0.440 and 0.511, respectively.

Bullough et al. [15] developed another SVM based model for predicting software exploitation with feature dataset built from open sources such as the NVD, exploit DB and Twitter. They conducted several experiments to evaluate the impact of existing challenges such class imbalance, temporal intermixing on effective prediction. They claimed that the NDV database and social media alone are unlikely to have enough predictive capability in creating a usable and realistic exploitation model, mainly because of the prevailing class imbalance in the dataset. Their best model achieved the accuracy, precision, recall and F1 score of 0.909, 0.519, 0.334 and 0.406, respectively, but provided no information on how to address the class imbalance. The results clearly showed overfitting to the major class.

Queiroz et al. [16] developed another SVM-based vulnerability exploitation prediction model using Twitter posts (tweets). This model achieved accuracy, precision, recall and F1 score values of 0.94, 0.68, 0.46 and 0.55, respectively. Their performance metrics showed overfitting to the majority class.

Sabottke et al. [3] developed another twitter-fed data-based vulnerability exploitation prediction model with the SVM algorithm. The ground truth was taken from ExploitDB, OSVDB, Microsoft security advisories and the descriptions of Symantec’s anti-virus and intrusion protection signatures. This research attempted to assess the opportunities for early detection of exploits for vulnerabilities in the presence of both benign and adversarial noise posts in tweets. Their model achieved precision and recall values of 0.80 and 0.10, respectively, showing a clear overfitting to the major class.

Jacobs et al. [17] tested and evaluated three practical vulnerability remediation strategies using CVSS scores, published exploits and vulnerability attributes. Multiple commercial datasets from FortiGuard Labs, SANS Internet Storm Center, Secureworks CTU, Alienvault’s OSSIM metadata and the Reversing Labs were merged with NVD to create a feature set. They utilized SVM algorithm to model a series of vulnerability remediation strategies and to compare the performance of each with regard to coverage and efficiency. Their best model achieved accuracy, efficiency (precision) and coverage (recall) scores of 94.5%, 71.4% and 37.3%, respectively, for the highly balanced strategy. The model result showed a clear overfitting to the major class.

Zhang et al. [18] proposed a new feature called the “TTNV” (Time to Next Vulnerability) and experimented with various classification algorithms to predict the time to the next vulnerability for a given software application. They utilized time and version attributes as features from the NVD dataset to experiment with Linux kernel, Microsoft Windows, IE and Mozilla to predict the TTNV and concluded that it was difficult to build good prediction models based on the data available in the NVD. Their best model with a Linux kernel achieved low values for TPR (True Positive Rate), FPR (False Positive Rate) and accuracy of 65.8%, 59.5% and 61.8%, respectively.

Bozorgi et al. [19] developed a linear SVM-based vulnerability exploitation prediction model and also attempted to predict the timeframe for the expected exploitation. They utilized NVD and OSVDB as feature set and exploit DB along with a few private datasets as the ground truth. Two classifier models, such as “offline” and “simulated online” were developed for the prediction. The accuracy for offline and simulated online models achieved values of 0.89 and 0.51, respectively, for the seven-day timeframe of exploitation probability.

### 2.2. Limitations of Existing Models

A good exploitability prediction model should have good overall performance metrics values for accuracy, precision and recall to be practically usable. Among the vulnerability exploitation prediction models from existing research, only the one by Edkrantz et al. [13] has good overall performance values for accuracy, recall and F1 score of 0.83, 0.82 and 0.83, respectively, but this model was developed in 2015 and did not reproduced the same results with recent NVD data in our test. . Moreover, they did not mention any measure to address the existing class imbalance. Other models suffer from poor overall performance values, almost all of them have good accuracy but poor precision, recall and F1 score performance. Models with high accuracy yet poor precision and recall or F1-score suggest that they have been over-fitted with the majority class and indicate unsuccessful addressing of the existing class imbalance in the target class since the number of exploited vulnerabilities is much less than the number of unexploited vulnerabilities. Existing research works have tried to address the issue of the existing class imbalance by using synthetic data, resampling and algorithmic approaches instead of developing a cost function in which the data are not artificially resampled. These approaches can improve the performance of models to some extent but cannot address the root cause of the existing class imbalance causing over-fitted models in favour of the major class. Besides that, some models have utilized social media data which are sometimes intentionally polluted by malicious parties and may also bring information about exploits (ground truth) in prediction features since social media and blogs would mostly discuss on exploited vulnerabilities.

## 3. Vulnerability Data Preparation

The vulnerability data were aggregated from three sources: the National Vulnerability Database (NVD) [20], exploit dB [21] and Symantec attack signatures [22], and then merged using the Common Vulnerabilities and Exposures Identifier (CVE_ID) as the key. The open-access “cve-search” GitHub project [23], which is a tool to import CVE, CPE, and CWE data into MongoDB to facilitate the search, manipulation and processing of CVE data, was utilized in extracting the data from the NVD database. The advantage of the “cve_search” tool lies in its ability to avoid making direct and public lookups into the CVE databases. This tool was modified in extracting the required data for this research from the NVD repository to populate the data in MongoDB in 12 tables, of them the tables named “cve”, “cpe” and “cwe” contain the most important vulnerability data. A total of 146,183 unique CVE_IDs together with other feature fields, such as “cwe”, “cvss”, “references”, “impact”, “summary”, “Published”, “Modified” and so forth were populated for the years 1997–2020 in the “cve” table from the NVD database. The following is an example of the data row from the “cve” table in the json format (refer to Figure 1). The data from “cpe” and “capec” were then merged with the “cve” table on the “CVE_ID” key.

The Exploit DB (EDB) contains information and a proof-of-concept code of common exploits. Although the EDB contains codes for exploits with no official CVE-numbers, yet only exploits with official CVE_IDs were used in this research and were considered as the ground truth label (binary class). The NVD database contains Common Weakness Enumeration (CWE) numbers which are unique numbers for vulnerability categorization such as cross-site-scripting or OS command injection. A total of 1005 unique CWE-numbers were populated in the “cwe” table. Symantec attack signatures were extracted by custom-coded web scrapper and merged with exploit db on CVE_ID key.

In constructing the label vector *y* with true values, a vulnerability was considered as having an exploit (*yi* = 1) if it had a link to some known exploit from the CVE-numbers scraped from the merged series of the EDB and Symantec antivirus signatures. A total of 11,720 labels were found with proof of exploits (y = 1). In order not to build the answer *y* into the feature matrix *X*, extracted links in the populated database containing information about the exploits have been excluded.

In this study, three new attributes have also been intuitively created for each vulnerability (CVE_ID/id) from feature fields of the “cve” table (refer to Figure 2). These novel features were utilized for the base models together with the processed textual data to examine if features other than the text data have any significant impact in classification modelling:“config” which is the size of the nested documents in the vulnerable_configuration_cpe_2_2 field;“Days_diff” which is the difference between the Modified and Published dates and converted into number of days;“Ref” which is the size of nested documents in the reference field.

The text-based feature called “summary” has undergone various pre-processing steps, such as the conversion into lowercase, removal of HTML tags, removal of punctuation marks, removal of stop words [English], stemming, and so forth. An example of the final feature data can be seen in Figure 3. Social media data have not been included as text feature in this study because, in many cases, social media contain intentionally manipulated conversations to create polluted data. Besides that, most discussions contain information about vulnerabilities which have already been or are about to be exploited, therefore adding these texts will bring label class information into the prediction class.

## 4. Approaches to Address Class Imbalance

### 4.1. Design of Cost Function

Most machine learning algorithms perform well when the number of instances of each class is roughly equal. When the number of instances of one class far exceeds the other, the data suffer from the class imbalance problem by having a low predictive accuracy for the infrequent class (having a low precision value). This, therefore, causes the machine learning model to become practically unusable and unreliable. This is a surprisingly common problem in machine learning (specifically in classification) such as with the extracted NDV dataset (a ratio of 1 to 12 in favour of the major class) used for this research (refer to Figure 4), where 0 = majority class which is “not exploited”, 1 = minority class which is “exploited”). Although previous research works have relied on data and algorithmic approaches on the imbalanced data, it has been observed that no feature engineering step can help fix the real problem of class imbalance when it is severe. However, they help in dealing with other issues such as high dimensionality and overfitting. The approach used in this study differs from the previous works since a new cost function was designed to address the class imbalance without artificially changing any data which happens secretly inside the algorithmic computations.

In this research work, an appropriate cost function which is the most advanced option in addressing class imbalance was designed out of the existing attribute feature (cwe_value) to work as an error mitigation weighting feature. It can be applied to the examples in the minority class in large value and to the majority class in small value, and can therefore address the class imbalance problem naturally without altering or artificially resampling the dataset when creating a classification model. This will naturally generalize the favour for the rare class in computation. Therefore, the cost function will assign a large weight to examples from the minority class while a small weight is assigned to examples from the majority class (refer Figure 5 for the conceptual representation).

A feature called the “cwe_value” was extracted from the NVD database and later merged with the “cve” table on the “CVE_ID” key. This feature was utilized in designing the cost function which is named “co_efficient_balance”. Initially, the “cwe_value” feature was grouped while the “CVE_ID” was aggregated according to its size (count). A new feature column called the “cwe_total_number” was created by aggregating according to the “CVE_ID” size for each “cwe_value” and was divided according to the respective class size. The ratio between the classes was then calculated. If the ratio is more than 5, the classes would thus be considered as highly imbalanced. The co_efficient_balance was then calculated as “*float (np.arange(1.0,0.1,(majority class/minority class)/2))*”. The first value should start from 1.0 and the trail and optimization would continue with a step value of 0.1. The final value of the “co_efficient_balance” was calculated and selected such that it can produce the best fit for all performance metrics, such as accuracy, recall, precision, f-score and Score-AUC. The final selected value was multiplied to each respective row of the “cwe_total_number” feature column which was then utilized as the predictive feature together with the textual feature in the classification algorithm. The algorithm for calculating the cost function (co_efficient_balance) can be found in Figure 6.

### 4.2. Creation of Custom Trained Word Vectors

There are two options available in using the textual feature through word embedding, either by using pre-trained vectors or by using custom vectors, and the choice depends on how much data are available for the custom use case [24]. Since we have large textual data in the “claen_summary” field of the processed dataset, it was thus suitable to choose the custom word vectorization. This is because it would be very specific to the context of the corpus, since a word is understood by its context which means the words that come in the neighbourhood of a particular word. Therefore, a custom-trained word vector was created using the “gensim’” model with the “Word2Vec” technique of 5000 words by using textual data. “vulnerability.vec” was later utilized as the embedded layer feature to train the classification algorithms, particularly for the neural networks (*“gensim.models.Word2Vec(clean_summary, min_count = 5, size = 5000, workers = 4, window = 5*)”). The performance of this custom-trained word vector in comparison to the popular pre-trained word vectors, such as GloVe and wiki.300d.vec, was found better and is discussed in the next section.

## 5. Experiments, Results and Analysis

### 5.1. Effect of Existing Class Imbalance

Since the overwhelming number of examples from the majority class will dominate the performance measure for accuracy over the number of examples in the minority class, accuracy as a performance measure is inappropriate for the imbalanced classification problem. This means that even an unskilful model can achieve very high accuracy scores such as 0.90 or more, depending on the extent of the existing severity of the class imbalance while the precision value which represents the ability of the model to correctly classify the minority class will be low. Therefore, the alternative is to use other performance metrics together with accuracy, such as the precision, recall and F1-score metrics [25]. Precision is the ratio between the True Positives and all the Positives, so it represents the proportion of positive identifications that have been correctly predicted by the model. Recall, which is also known as Sensitivity, is the ratio between correctly predicted positive identifications to all observations in the actual class. Recall or Sensitivity therefore answers the question for the proportion of actual positives that have been correctly predicted by the model. The F1 Score is simply the harmonic mean of precision and recall (see Figure 7). The AUC provides the aggregated measure of the classification model performance across all possible classification thresholds. In all the metrics mentioned above, the value ranges between 0–1 and the higher the values, the better the model [26].

In the context of predicting most exploitable vulnerabilities with supervised classification-based machine learning models, the target label is a binary class in which the ratio between the number of exploited vulnerabilities (ground truth) and non-exploited vulnerabilities is approximately 1 to 12. Therefore, the number of exploited vulnerabilities is the minor class with a severe class imbalance in favor of the major class. The objective of this research is to better predict the positive (minor) class and so the performance metric of “accuracy” alone will be misleading for overall model performance. since a model can easily achieve a very high accuracy by predicting most of the major (negative) class, although the correct prediction of the desired (positive) class will be very low at the same time. Therefore, other performance metrics such as precision, recall, F1 score and AUC score have been considered alongside accuracy in order to analyse the model performance.

### 5.2. Development of Base Models

We have run the models in a Dell ZBook laptop with Intel(R) Core(TM) i7-5500U CPU @ 2.40 GHz and 16 GB RAM. In addition to using train/test split, 10-fold cross-validation utilized, so that all observations are used for both training and validation, and each observation is used once for validation to reduce sampling biasness. GridSearchCV technique was also utilized to optimize machine learning algorithms parameters which also supports validation process.

At first, base models were created with text data as the feature vectors processed by various advanced natural language processing techniques (refer to Table 1). The base models were created using different popular traditional as well as advanced neural networks for classification (refer to Table 1).

Models with deep neural networks were created only with text embedding as the feature through an added layer. This is because, unlike other NLP techniques, text features through word embedding tend to perform better with neural networks when there are enough data, such as in this research work. The base model performance metrics are shown in Table 2, Table 3 and Table 4.

Another base model was also created with novel created features, such as “Ref”, “days_diff” and “config” together with textual features such as TF-IDF at N-gram level using the popular XGboost algorithm which is a top performing base model for existing imbalance classes. However, there was no noticeable improvement in the performance metrics with 0.94, 0.48, 0.78 and 0.59 for accuracy, precision, recall and F1-score, respectively. These result metrics therefore conformed to previous research [15] findings that other integer-based features, when considered along with the textual feature, do not noticeably improve the performance of the model and the textual feature plays the principle predictive role in classification. Although the base models have good accuracy, yet these results are misleading. The model acts like a Zero Rule model where only the majority class has been discovered, while the rare class that is more interesting in this research containing the positive class has largely been ignored. The classification algorithms have been overwhelmed by the large class resulting into misclassification because their output is simply the most frequent class in the dataset. The algorithms have clearly provided more weightage to the major class in computing the accuracy and, thereby, suffer from the over-fitting issue due to the existing class imbalance. Therefore, other performance metrics, such as precision, recall and F-score were found to be very low. These models cannot be used in the real case scenario and the root problem that the classification algorithms could not overcome is the existence of the highly imbalanced label classes.

### 5.3. Models with Data Manipulation and Algorithmic Approaches

To overcome this problem and to improve the base models, experiments were performed on various data and algorithmic approaches to handle the imbalanced data. Two algorithms, namely the “Resampling Technique with different ratios Random (SMOTE)” with Linear Regression and the Random Over-sampling with Long Short-term Memory Network, a variant of the recurrent neural network, were utilized for the Data Level approach. The Bagging-based technique such as Random Forest with “weight = balanced” parameter was utilized for the algorithmic-based ensemble technique. The Boosting-Based technique was experimented with the XG Boost algorithm and finally, the Cost-Sensitive Neural Network with parameter “weights = 1:12” was utilized with keras API framework since the majority class is 12 times greater than the majority class. Table 5 shows the results for the model performance for data and algorithmic approaches for imbalanced data.

When comparing the results between two sets of base models (Table 5 vs. Table 1), it can be observed that only a slight improvement has been achieved by the models in Table 5 from the models in Table 1, specifically for the F1-score. Only the Cost-Sensitive Neural Network has shown a significant improvement for the precision result but is still insufficient to be considered for real life usage**.** Therefore, models developed artificial resampling of data through data level and algorithmic approaches to address the existing high-class imbalance have achieved high accuracy, similar to the base models in Table 1 but failed to achieve the desired overall performance scores (F-score, precision and recall). Therefore, the base models in Table 5 still suffer from the overfitting problem for the majority class (high accuracy and low precision).

### 5.4. Models with Data Manipulation and Algorithmic Approaches

These final models were developed using integer attributes derived through a novel cost function and a word embedded layer in the neural network derived through a custom-trained word vector. The scope of the novel-designed cost function (co_efficient_balance) is limited to the dataset prepared by a union of the national vulnerability database, Exploit DB and Symantec attack signatures on CVE_ID key. The value of the cost function was calculated using the proposed algorithm mentioned in Section 2. If Y2 and Y1 are the sizes (value_counts) of the label classes (binary), the values of Y2 and Y1, in this study, are 135,299 and 11,721, respectively. Since y2 > y1, the variable Y3 can be calculated as “*round (y2/y1)*” which is 12, therefore y3 > 5 which means that the class imbalance is high. Now, the “co_efficient_balance” can be calculated using the function “*float (np.arange(1, 0.1, 12/2))*”. The “co_efficient_balance” can have the value of 1~6 ((Y2/Y1)/2) with a step value of 0.1 and is calculated using the trail and optimization for the best fit to achieve the best overall performance metric for the classification model. The trail and optimization process should start with the value 1 and can be increased with the step value of 0.1 in a loop to run the models to achieve the best fit performance metrics. The value for “co_efficient_balance” has been optimized at 4.8 for the extracted dataset in this research as the best fit for the overall model performance. This value of “co_efficient_balance” was then multiplied to each row of the feature column “cwe_tot_number” and the feature was then renamed to something more meaningful such as “class_balance”, to be utilized as the predictive feature together with the text features features for classification algorithms. Two classification algorithms, namely the Convolutional Neural Network (CNN) and the Long Short-term Neural Network (RNN-LSTM) that performed well for this classification job in base models, were considered for the final run with “class_balance” and word embedding through token-embedded mapping with pre-custom-trained word vectors as prediction features for classification. The results from these runs, together with the best results from the existing previous research, can be seen in Table 6.

The performance metrics of the final classification have much better overall values than the results of the existing similar research. For example, the classification model with RNN-LSTM achieved the best performance metrics with accuracy, precision, recall and F1-Score values of 0.92, 0.89, 0.98 and 0.94, respectively, which demonstrates very good overall performance and has been least affected by the overfitting problem, if any. The improvements of the base models are highly significant, especially in precision, recall and F1-score values considering the existing class imbalance in the dataset. Therefore, the novel feature “class_balance” powered by the cost function (co_efficient_balance’) was truly capable of addressing the class imbalance problem in the classification computation. The ROC (receiver operating characteristic**)** curve is another performance measurement metric for classification models at various threshold settings. The ROC is a probabilistic curve while the AUC measures the degree of separation for classification classes. This therefore demonstrates how much a classification model can distinguish among classes and, therefore, the higher the AUC, the better the model is at defining class separation [26]. Our final model has achieved the AUC score of 0.97 in the ROC curve (refer to Figure 8). Besides that, the confusion matrix has shown highly accurate results (refer to Figure 9 with 33833 correct predictions in the negative class (TN) with 0 incorrect prediction (FN) and 2428 correct predictions in the positive class (TP) with only 394 incorrect predictions in the test data set (FP).

The loss function for classification in the RNN-LSTM network against the epoch (3 epochs for example) shows the high similarity between training and the test class (refer to Figure 10) computation. Therefore, the model is nicely fitted both for training and test datasets. This result also promises a strong ability to perform well when the model is applied to a new dataset (test class).

A special ANN called the Recurrent Neural Network (RNN) is designed to utilize loops to persist information from knowledge learnt from past data points. This is achieved by maintaining memory cells to capture information from past data points in the sequence (refer to Figure 3) and, therefore, the RNN models would usually perform better than the NNMs in terms of modelling dependencies between two points. However, one significant drawback of the RNN is that when the data sequence starts to increase, the network tends to lose information of the historical context over time. A variant of the RNN called the Long Short-Term Memory Network (LSTM) can solve this problem since it maintains cells to keep information for the previous nodes irrespective of the sequence size. The LSTM network can achieve this by maintaining three or four gates, such as the input, output and forget gates (refer to Figure 11) [27]. Therefore, the RNN-LSTM network, aided by a large dataset, has performed better for the models developed in this research. The model performance results have also been examined by comparing the custom-trained word vector (vulnerability.vec) and the two popular pre-trained word vectors such as GloVe (Global Vectors for Word Representation) and wiki-news-300d-1M.vec (English). As expected, the custom-trained word vector performed much better with the CNN and RNN-LSTM networks (refer to Table 7), specifically with the precision score. Two factors contributed to this performance improvement. Firstly, the text corpus in the dataset was large enough to create and train the classification models efficiently. Secondly, as expected, the custom-trained word vectors created out of the local data were able to capture the context of word neighbourhood better than the popular pre-trained word vectors.

Four top vendors by number of published vulnerabilities in cve-details [28] were selected to test the developed models. A custom-coded data extractor was utilized to dump data for vendors, such as Microsoft, HP, Oracle and IBM. These data were engineered accordingly and optimized values for co_efficient_balance were measures according to the cost function algorithm. As expected, the models performed very well (see Table 8) for the test datasets promising reliable usage in practical case scenarios.

## 6. Conclusions

In this research work, the performance metrics of the recent research findings on predicting vulnerabilities that are most likely to be exploited with classification machine learning modelling have further been improved through the use of two features. Firstly, the creation of a novel cost function (termed co_efficient_balance) which is the most advanced method for addressing the existing class imbalance. Secondly, the creation of a custom-trained word vector with 5000 words from a large text corpus in the “summary” field of the NVD dataset, to be utilized as word embedded through an added layer in the deep neural network. The utilized dataset from the “National Vulnerability Database” is the most comprehensive database for vulnerability data and has the largest user-base. The best performing model of this study with RNN-LSTM has achieved accuracy, precision, recall, F1-Score and AUC values of 0.92, 0.89, 0.98, 0.94 and 0.97, respectively, and has easily outperformed any other model in previous research works involving the NVD dataset. The overall high performance score also indicates that the developed model has been least affected by the overfitting issue. The custom-trained Gensim-based word vector (termed as “vulnerability.vec”) also significantly outperformed popular pre-trained word vectors such as “GloVe (FastText)” and “wiki-news-300d-1M.vec (Word2vec)”. The long short-term neural network, a variant of the recurrent neural network, performed best among all algorithms for this classification problem since it could handle the large dataset and was able to effectively remember past data nodes in the memory cells for future calculation. The developed model took significantly shorter time to complete the computation compared to the base models developed with data and algorithmic approaches. For example, the final model with the RNN-LSTM took only 399 s and 98 ms to complete the calculation whereas the base model with the SMOTE technique took 7234 s and 233 ms. Therefore, our developed vulnerability exploitation prediction model will be usable for any organization that has limited computational resources in its premises or in a cloud environment.

## Figures and Tables

**Figure 1 sensors-21-04220-f001:**
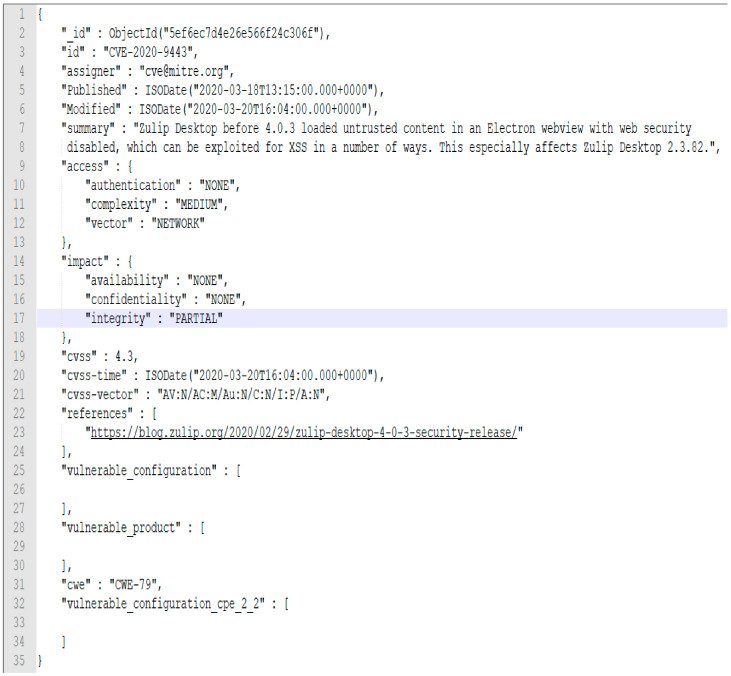
Features of the “cve” table.

**Figure 2 sensors-21-04220-f002:**
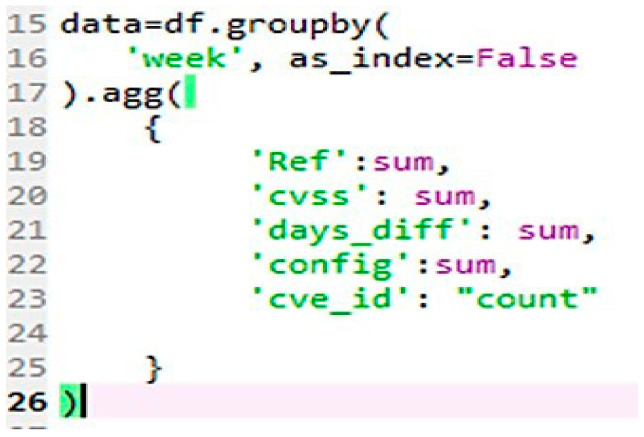
Creation of features.

**Figure 3 sensors-21-04220-f003:**
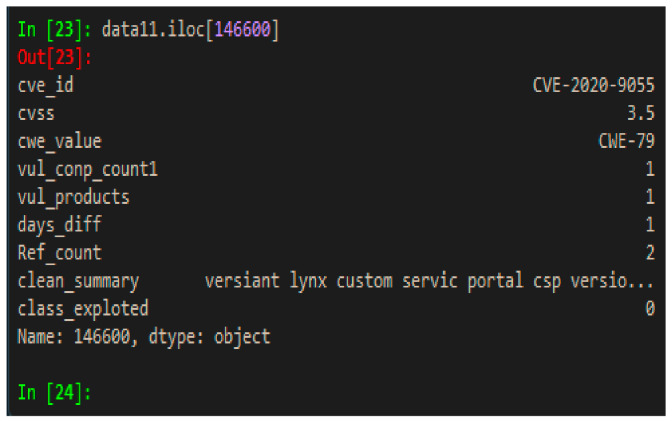
Final data features.

**Figure 4 sensors-21-04220-f004:**
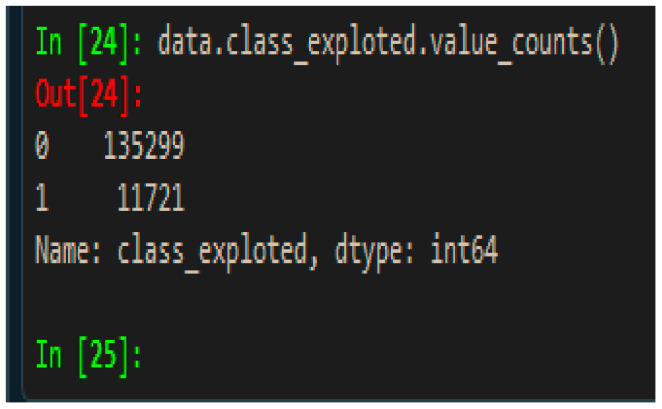
Existing class imbalance.

**Figure 5 sensors-21-04220-f005:**
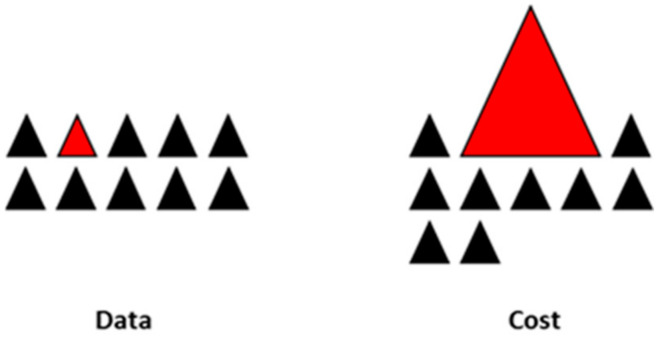
Conception of cost function.

**Figure 6 sensors-21-04220-f006:**
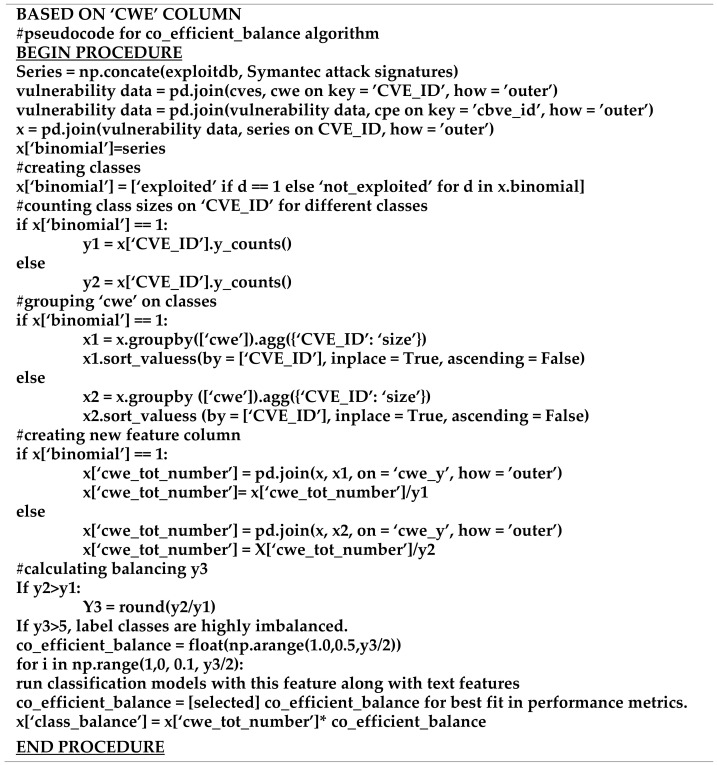
Algorithm (pseudocode) for the cost function.

**Figure 7 sensors-21-04220-f007:**
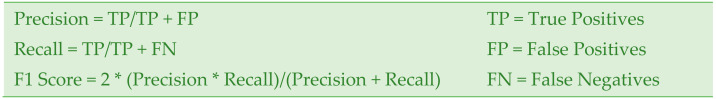
Performance metrics.

**Figure 8 sensors-21-04220-f008:**
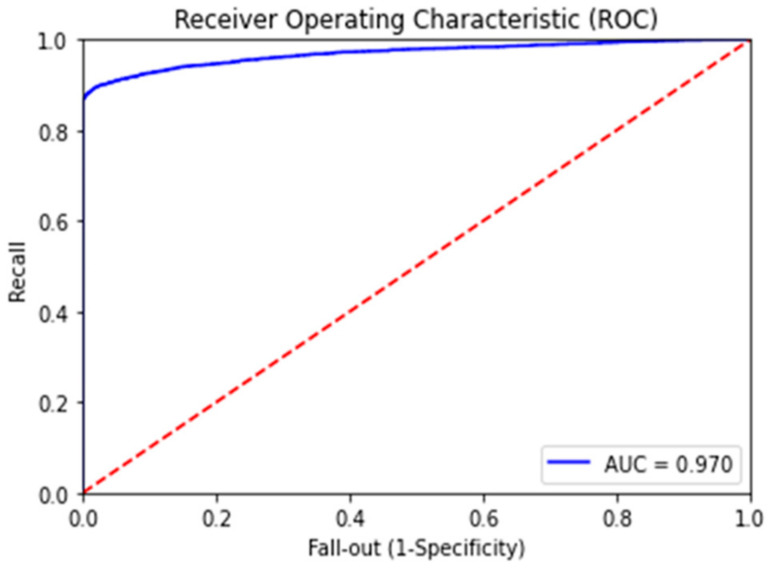
ROC curve.

**Figure 9 sensors-21-04220-f009:**
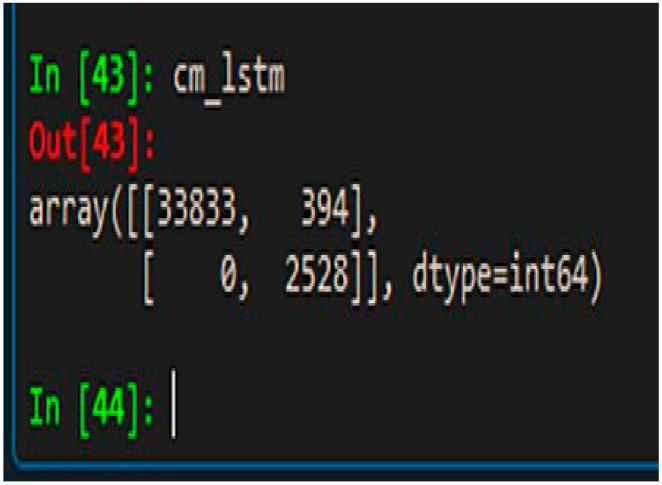
Confusion matrix.

**Figure 10 sensors-21-04220-f010:**
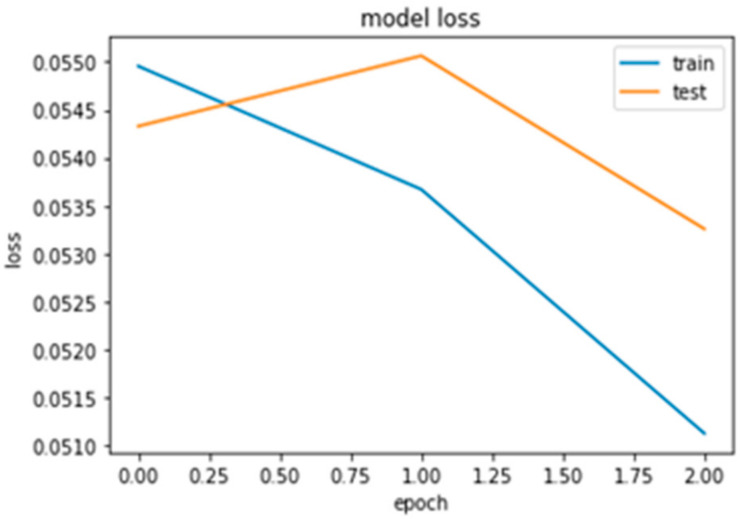
Loss function comparison.

**Figure 11 sensors-21-04220-f011:**
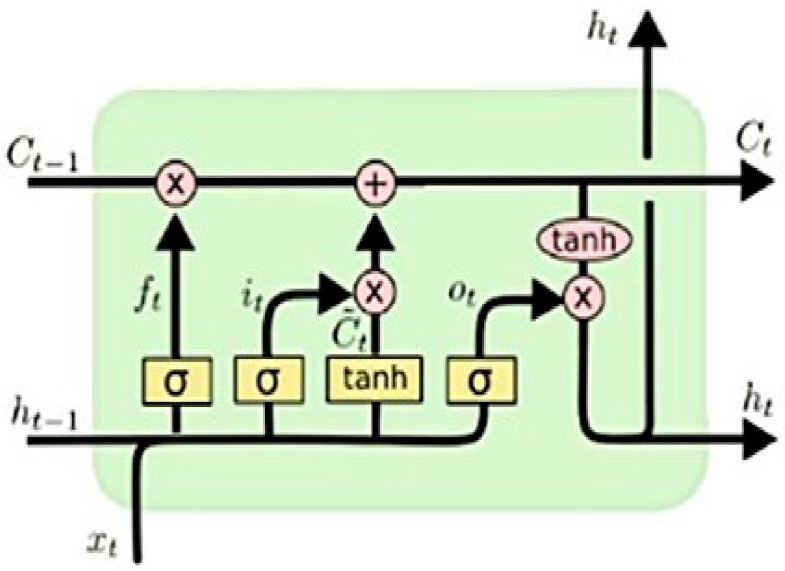
An LSTM node (Source: [27]).

**Table 1 sensors-21-04220-t001:** NLP processing and base model algorithms.

NLP Processing Techniques	Classification Algorithms
Count Vectors as features TF-IDF Vectors as features Word level N-Gram level Character level Word Embeddings as features Topic Models as features	Naive Bayes Classifier Linear Classifier Support Vector Machine Bagging Models (RF) Boosting Models (XGBoost) Deep Neural Networks Convolutional Neural Network (CNN) Long Short-Term Model (LSTM) Gated Recurrent Unit (GRU) Bidirectional RNN Recurrent Convolutional Neural Network

**Table 2 sensors-21-04220-t002:** Base models 1.

	Features	Count Vectors				Word Level TF IDF Vectors				Ngram Level TF IDF Vectors			
Models		Accuracy	Precision	Recall	F1-Score	Accuracy	Precision	Recall	F1-Score	Accuracy	Precision	Recall	F1-Score
Naive-bayes		0.89	0.57	0.38	0.46	0.92	0.57	0.38	0.46	0.90	0.57	0.38	0.46
Logistic Regression		0.93	0.61	0.61	0.45	0.93	0.29	0.68	0.40	0.93	0.33	0.69	0.44
SVM		0.92	0.35	0.61	0.43	0.93	0.35	0.60	0.41	0.93	0.34	0.61	0.42
Bagging Model(RF)		0.94	0.39	0.78	0.52	0.94	0.41	0.74	0.53	0.93	0.41	0.73	0.53
Boosting Model (XGBoosting)		0.93	0.39	0.70	0.50	0.93	0.35	0.72	0.47	0.93	0.35	0.72	0.47

**Table 3 sensors-21-04220-t003:** Base models 2.

	Features	Topic Models as Features				Word Embeddings as Features			
Models		Accuracy	Precision	Recall	F1-Score	Accuracy	Precision	Recall	F1-Score
Naive-bayes		0.91	0.57	0.38	0.46	0.92	0.59	0.38	0.47
Logistic Regression		0.93	0.29	0.68	0.40	0.93	0.33	0.60	0.45
SVM		0.94	0.35	0.60	0.41	0.94	0.35	0.60	0.41
Bagging Model(RF)		0.94	0.37	0.77	0.50	0.94	0.37	0.77	0.50
Boosting Model (XGBoosting)		0.94	0.39	0.71	0.51	0.94	0.40	0.72	0.53

**Table 4 sensors-21-04220-t004:** Base models 3.

	Features	Word Embeddings as Features			
Models		Accuracy	Precision	Recall	F1-Score
Convolutional Neural Network		0.92	0.32	0.66	0.43
Recurrent Neural Network LSTM		0.92	0.33	0.68	0.44
Recurrent Neural Network –GRU		0.92	0.26	0.69	0.38
Bidirectional RNN		0.92	0.34	0.68	0.45
Recurrent Convolutional Neural Network		0.92	0.20	0.73	0.31

**Table 5 sensors-21-04220-t005:** Model performance for data and algorithmic approaches.

	Features	Ngram Level TF IDF Vectors				Word Embeddings as Features			
Models		Accuracy	Precision	Recall	F1-Score	Accuracy	Precision	Recall	F1-Score
Random Over-sampling (RNN-LSTM)		0.50	0.69	0.68	0.64	0.52	0.70	0.68	0.64
SMOTE (LR)		0.89	0.40	0.69	0.48	0.90	0.44	0.69	0.55
Bagging Model(RF)		0.93	0.49	0.62	0.55	0.93	0.50	0.68	0.59
XG Boost		0.94	0.39	0.78	0.52	0.94	0.41	0.74	0.53
Boosting Model		0.93	0.39	0.70	0.50	0.93	0.35	0.72	0.47
Cost-Sensitive Neural Network		0.91	0.77	0.29	0.42	0.92	0.78	0.31	0.55

**Table 6 sensors-21-04220-t006:** Comparison of the model results.

ML Models	Our Model (RNN-LSTM)				Model in [ [14] ] (LibSVM)				Model in [ [18] ] (SVM)			
Features	Word Embeddings				TF-IDF				TF-IDF			
Performance Metrics	Accuracy	Precision	Recall	F1-Score	Accuracy	Precision	Recall	F1-Score	Accuracy	Precision	Recall	F1-Score
Performance Values	0.92	0.89	0.98	0.94	0.83	0.84	0.82	0.83	0.94	0.71	0.37	0.54

**Table 7 sensors-21-04220-t007:** Comparison of word vectors.

Word Vectors		Vulnerability.Vec				Glove.Vec				Wiki-News-300d-1M.vec			
	Features	Word Embedding				Word Embedding				Word Embedding			
Models		Accuracy	Precision	Recall	F1-Score	Accuracy	Precision	Recall	F1-Score	Accuracy	Precision	Recall	F1-Score
(RNN-LSTM)		0.92	0.89	0.98	0.94	0.92	0.64	0.99	0.78	0.92	0.54	0.91	0.68
CNN		0.92	0.67	0.93	0.78	0.92	0.36	0.75	0.55	0.92	0.38	0.77	0.51

**Table 8 sensors-21-04220-t008:** Model performance for test datasets.

Test Dataset	Model Algorithm	Target Class Ratio	Co_Efficient_Balance(Optimized Value)	[Accuracy, Precision, Recall, F1-Score, AUC-Score]
Microsoft	RNN(LSTM)	1:12	3.6	0.98, 0.88, 1.00, 0.94, 0.94
HP	Naïve-Bayes	1:16	7.8	0.98, 0.98, 0.98, 0.98, 0.94
Oracle	Linear Regression	1:43	20.7	0.82, 0.75, 0.84, 0.79, 0.81
IBM	Naïve-Bayes	1:64	30.9	0.83, 0.85, 0.83, 0.76, 0.79

## Data Availability

Not applicable.

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
