# Peer review of "An Improved Vulnerability Exploitation Prediction Model with Novel Cost Function and Custom Trained Word Vector Embedding"

_sensors, 2021, doi:10.3390/s21124220_

Round 1
Reviewer 1 Report
The paper is interesting and well written. In my opinion the topic is relevant and the proposal sound. Conclusions are supported by the experiments and results. In general, I think the paper may be accepted as it
Reviewer 2 Report
The authors of this paper focused on a very important area of cybersecurity, that of the prediction of the exploitation of vulnerabilities by malicious parties. This type of work corresponds to a wide range of computational infrastructure with the Cloud ecosystem, being the number one candidate to adopt it. The authors have created a novel cost function and a custom-trained word vector that were combined in a prediction model with very good results.
This is a very interesting paper. However, the reviewer would like to raise some pointers to the authors.
The paper needs proofreading as there are grammar and syntax mistakes spread throughout the manuscript.
Some citations are broken (e.g. [21] appears twice)
Figures 2,10 and 11 need better resolution or to be converted in another form (e.g. algorithm)
A table at the end of the related work would help to summarize how the specific approach is superior (with tick boxes).
The authors should detail the hardware details of their testbed environment.
The performance metrics should be explained in the context of the exploitation prediction.
Reviewer 3 Report
The study presents a machine learning approach to predict exploitation of vulnerabilities. Several different machine learing setups are compared. A novel mechanism presented in a paper is a cost function.
The study carefully discusses details of the algorithms and makes a nice contribution. However, it does not exactly do what title and abstract promise. The main contribution is the experimental comparison of various algorithms to predict exploitation of vulnerabilities. It does not provide information inhowfar the results would be good enough to be useful in practice. An evaluation according to business cases and real numbers is missing - which is a pity. Also lots of information about the data set and the experimental validation are missing. E.g., references to the algorithms used. is the data set publicly available or the experimental platform. For a journal publication I would like to be able to conduct experiments myself and learn more about the data sets and the experiments.
Adding references, more transparency, changing title and intro such that this matches the main contribution - this needs to be done and then the paper will make a nice contribution.
Round 2
Reviewer 2 Report
The authors have addressed all the comments.
This manuscript is a resubmission of an earlier submission. The following is a list of the peer review reports and author responses from that submission.